Oropharyngeal microbiome dysbiosis in esophageal squamous cell carcinoma: taxonomic shifts, metabolic reprogramming, and geographic disparities in a high-incidence cohort

Liu Ying 1
Wu Erman 2 3
Cheng Fang 1
Zhang Meng 1
Rou Qian 1
Nuertai Zinati 1
Xu Maorong 1
Xu Shanshan 1
Li Minghui 1
Zhang Lei 1
Nasiroula Aheli Ahl1602@163.com 1
1 Special Needs Comprehensive Department, Affiliated Tumor Hospital of Xinjiang Medical University, Xinjiang Medical University , Urumqi , China
2 Department of Neurosurgery, The First Affiliated Hospital of Xinjiang Medical University, Xinjiang Medical University , Urumqi , China
3 Department of Computer Science and Information Technologies, University of A Coruña , A Coruña , Spain
Uversky Vladimir
Electronic publication date: 2025 Oct 6
Publication date: 2025
Volume: 13
Electronic Location ID: e20009
Received 2025 Apr 9; Accepted 2025 Aug 7
Copyright: ©2025 Liu et al.
Copyright year: 2025
Copyright holder: Liu et al.
License: This is an open access article distributed under the terms of the Creative Commons Attribution License, which permits unrestricted use, distribution, reproduction and adaptation in any medium and for any purpose provided that it is properly attributed. For attribution, the original author(s), title, publication source (PeerJ) and either DOI or URL of the article must be cited.
License URL: https://creativecommons.org/licenses/by/4.0/

Keywords: Esophageal cancer, Oral microbiota, 16S rRNA, Bioinformatics, Throat swab

Funding: State Key Laboratory of Pathogenesis, Prevention and Treatment of High Incidence Diseases in Central Asia, Xinjiang Medical University SKL-HIDCA-2022-SG5 The work was supported by the State Key Laboratory of Pathogenesis, Prevention and Treatment of High Incidence Diseases in Central Asia, Xinjiang Medical University (SKL-HIDCA-2022-SG5). The funders had no role in study design, data collection and analysis, decision to publish, or preparation of the manuscript.

==============================
Background

Esophageal squamous cell carcinoma (ESCC) is a leading cause of cancer mortality globally, with pronounced geographic disparities in incidence. Emerging evidence links oral microbiome dysbiosis to ESCC pathogenesis, yet comprehensive insights into microbial diversity, taxonomic shifts, and functional alterations in high-risk populations remain limited.

Methods

Using 16S rRNA amplicon sequencing, we compared the oral microbiome of ESCC patients and healthy controls from a high-incidence region in Northwest China. Alpha and beta diversity metrics, taxonomic composition, and predicted functional pathways were analyzed to identify microbial signatures associated with ESCC.

Results

ESCC patients exhibited significantly elevated microbial richness (observed amplicon sequence variants (ASVs), Chao1, ACE; p < 0.05) but comparable Shannon/Simpson diversity to controls. Unique amplicon sequence variants (ASVs) were more prevalent in ESCC samples, and principal component analysis confirmed distinct community structures (p < 0.05). Taxonomically, Streptococcus and Neisseria dominated both groups, but ESCC patients showed enrichment of Gemella (p = 0.0003) and Corynebacterium (p < 0.00001), alongside depletion of Prevotella_7 (p = 0.0002) and Moraxella (p < 0.001). Functional profiling revealed upregulated amino acid metabolism (e.g., beta-alanine and valine degradation) and downregulated carbohydrate metabolism in ESCC-associated microbiota.

Conclusion

This study uncovers unique oral microbial signatures in ESCC patients from a high-incidence region, characterized by increased richness, taxon-specific shifts, and metabolic reprogramming favoring amino acid catabolism. These findings highlight the potential of microbial biomarkers for ESCC detection and provide mechanistic insights into microbiome-driven carcinogenesis. The geographic specificity of the cohort underscores the urgency of tailored interventions in high-risk populations and advances our understanding of microbial contributions to esophageal cancer.

Introduction

Esophageal squamous cell carcinoma (ESCC) ranks as the sixth leading cause of cancer-related mortality globally, with over half of all cases concentrated in Asia, particularly in China (Chen et al., 2023; Abnet, Arnold & Wei, 2018; Zhu et al., 2023). This malignancy exhibits striking geographic disparities, with incidence rates in high-risk regions such as Northwest China being markedly elevated compared to global averages (Abnet, Arnold & Wei, 2018; Guo et al., 2017). While established risk factors—including tobacco use, alcohol consumption, dietary carcinogens (e.g., pickled vegetables, hot beverages), and socioeconomic determinants—contribute to ESCC pathogenesis (Yokoyama et al., 2019; Thrumurthy et al., 2019; Conway, Wu & Tian, 2023; Sheikh et al., 2019), emerging evidence implicates the oral microbiome as a critical modulator of disease development and progression (Carter et al., 2022; Asili et al., 2023; Lv et al., 2019).

The oral cavity serves as a reservoir for esophageal microbiota, with dysbiosis increasingly linked to gastrointestinal carcinogenesis (Yin et al., 2020; Chen et al., 2015). Dysbiosis in the oral cavity and pharynx can potentially trigger carcinogenesis. It does so via multiple pathways: by inducing chronic inflammation, generating bacterial metabolites such as acetaldehyde, and causing direct damage to DNA. Certain pathobionts, for example, Porphyromonas gingivalis, play a role in promoting tumor development. They can stimulate the production of pro-inflammatory cytokines, like IL-6, and suppress the process of apoptosis. This combination of effects creates a favorable environment for tumors to progress (Devaraja & Aggarwal, 2025). In healthy individuals, Streptococcus dominates the esophageal microbiome, maintaining structural consistency across esophageal regions (Peters et al., 2017). However, ESCC patients exhibit marked taxonomic shifts, including enrichment of pathobionts such as Porphyromonas gingivalis, which promotes tumorigenesis via interleukin-6 (IL-6)-mediated inflammation in preclinical models (Wang et al., 2019; Zhao et al., 2020). Despite these advances, comprehensive analyses of oral microbial diversity, taxonomic composition, and functional dynamics in high-risk populations remain scarce, particularly in regions bearing disproportionate disease burdens.

Notably, prior studies have focused predominantly on low-incidence cohorts or generalized populations, overlooking the unique microbial and environmental interplay in understudied high-risk regions. This gap hinders the development of targeted prevention strategies and geographically relevant diagnostic tools. Furthermore, while microbial richness and metabolic reprogramming have been proposed as hallmarks of cancer-associated dysbiosis (Snider et al., 2019; NCCN, 2020), their specific roles in ESCC pathogenesis within high-incidence settings are poorly characterized.

Here, we conducted the first detailed investigation of oral microbiome alterations in ESCC patients from a high-incidence region in Northwest China using 16S rRNA amplicon sequencing. By analyzing alpha/beta diversity, taxonomic shifts, and predicted functional pathways, this study aims to (1) identify microbial signatures unique to ESCC in a high-risk population, (2) elucidate potential mechanisms linking dysbiosis to carcinogenesis, and (3) provide actionable insights for early detection and microbiome-targeted interventions. Our findings address critical gaps in understanding geographically driven microbial contributions to ESCC and underscore the urgency of tailored public health strategies in high-burden regions.

Materials and Methods

Study participants

A total of 30 patients with esophageal cancer in each stage were selected according to the NCCN Guidelines—Esophageal and Esophagogastric Junction Cancers (Version 1. 2020) (NCCN, 2020) and confirmed as squamous cell carcinoma by endoscopy and pathology (ESCC group, 16 men and 14 women). ESCC patients were recruited from the Affiliated Tumor Hospital of Xinjiang Medical University from March 2023 to August 2023. 31 healthy controls (control group, 16 men and 15 women) were recruited from the people who came to the hospital for physical examination at the same period. This study was approved by Ethics Committee of The Affiliated Cancer Hospital of Xinjiang Medical university (Ethical number: K-2024017).

The inclusion criteria and exclusion criteria were as follows:

Inclusion criteria: Age ≧18 years old, expected survival time ≧3 months, no history of surgery, chemotherapy, radiotherapy, and other local treatment, and complete mobility, in line with the relevant regulations of the ethics committee of our hospital, voluntarily signed the informed consent, and the compliance was good.

Exclusion criteria: (1) combined with other malignant diseases; (2) use of anti-inflammatory drugs, antibiotics, or immunosuppressants within the past 3 months; (3) professionally clean teeth within one month; (4) receiving neoadjuvant chemotherapy or radiotherapy; (5) communication difficulties; (6) difficulty in obtaining informed consent. Written informed consent was obtained from all participants before the study.

Throat swab sample collection: Throat swabs were collected ≥2 h post-prandial to minimize food residue interference. The swab was slowly placed behind the pharynx for about 5 s. Be careful not to scratch the tonsils. Immediately after scraping, the samples were vortexed in MoBiobuffer containing 750 ul. The sponge on the swab should be pressed on the wall of the centrifuge tube a few more times, about 20 s, to ensure transfer of the bacteria into buffer. The buffer was then frozen at −80 °C and stored until use.

DNA extraction and sequencing processing and analysis

Total genome DNA from samples was extracted using the QIAamp DNA Stool Mini Kit (Qiagen, Hiden Germany) combined with the bead-beating method. The DNA concentrations of each sample were adjusted to 50 ng/µl for subsequent 16S rDNA gene analysis. The bacterial DNA samples were stored at −80 °C for sequencing of the follow-up studies.

16S rDNA genes of V3–V4 region were amplified using universal primers, namely 341F (5′-CCTACGGGNGGCWGCAG-3′) and 805R (5′-GACTACHVGGGTATCTAATCC-3′). Then, we constructed the DNA library with the amplicons NEB Next Microbiome DNA Enrichment Kit (New England Biolabs. Inc, Ipswich, MA, USA), committed the PCR products to the MiSeq platform (Illumina, San Diego, CA, USA), and obtained the 250 (nt) paired end reads.

Quality control. All samples were processed under identical experimental conditions by the same experimenter, blinded to the sample status. Sample isolation and extraction were carried out in a sterile laminar flow hood, with all steps ensuring aseptic handling. All samples underwent sequencing in a single batch.

Data processing

We adopted QIIME2 software (version 2020.2.0) for the taxonomical annotation Qiime 2 (Bolyen et al., 2019). We obtained the amplicon sequence variants (ASVs) using deblur. Then, we aligned the ASVs to the feature classifier trained from the Green Genes Database and got the taxonomical annotation results. 16S rRNA gene-based microbial function enrichment prediction using PICRUSt2 (Douglas et al., 2020).

Statistical analysis

Using genus-level profiling, we calculated α-diversity and estimated β-diversity using Bray–Curtis distances between samples. Principal coordinate analysis (PCA) was performed based on Bray–Curtis distances using the R package ‘vegan’ (v2.6-4). Based on taxonomic profiling, the top ten most abundant genera were selected. Their abundances were compared between the ESCC group and the control group using the Wilcoxon rank-sum test, followed by Benjamini–Hochberg correction for multiple testing (adjusted p < 0.05). Six α-diversity indices were used to compare oral microbial diversity between the cancer group and the healthy group (Fig. 1A). Beta-diversity was estimated using Bray–Curtis distances. Clinical parameters (Table 1) were analyzed using independent t-tests (age) or chi-square tests (categorical variables) in SPSS v26.0.

Figure 1 The alpha diversity of oral microbiome was compared among the ESCC and healthy control groups.

(A) Boxplots comparing the alpha diversity of oral microbiota between ESCC and healthy controls, six alpha diversity indices are shown. (B) The Venn diagram shows the ASV overlap between the two groups. (C) PCA based on β diversity showed differences in the structure of the two microbial species in the two groups (p < 0.05).

Results

Baseline characteristics of participants

A total of 30 ESCC patients (16 male and 14 female) and 31 healthy controls (16 male and 15 female) were included in the study. Statistical analysis showed no significant differences in age, gender, smoking between the two groups (P > 0.05) (Table 1).

Alpha diversity analysis revealed distinct patterns between ESCC patients and healthy controls. Three richness estimators (Observed ASVs, Chao1 index, and ACE index) demonstrated significantly higher values in the ESCC group compared to controls (p < 0.05). While Shannon and Simpson diversity indices showed a moderate elevation in controls, no statistically significant intergroup differences were observed (p > 0.05). Notably, the Good’s coverage index was significantly higher in controls than in ESCC samples (p < 0.05) (Fig. 2A).

Table 1 Clinical characteristics of enrolled patients and healthy controls.

		Control (%)
n = 31	ESCC (%)
n = 30	P	
Age (years, mean ± SD)		65.77 ± 6.15	62.23 ± 9.00	0.077*	
Gender	Male	16 (51.61%)	16 (53.33%)	0.789#	
Female	15 (48.39%)	14 (46.67%)	
Smoking	Never	25	22	0.340#	
Ever	6	8	
Notes.

* Differences were detected using the independent sample T test.

# Differences were detected using the chi-square test. Thirty patients with ESCC (ESCC group, 16 males and 14 females) and 31 control subjects (control group, 16 males and 15 females) were enrolled in the study. Microbial Richness and Diversity.

Figure 2 Comparison of relative abundance between the two groups based on genus.

(A) Distance-based heatmap of pairs of two groups of samples at the genus level. (B) Top 10 significantly different species superposed bar graph with confidence intervals in two groups. (C) A bar plot of the relative abundance of the main bacterial taxa at genus for controls and ESCC groups.

Comparative analysis of amplicon sequence variants (ASVs) identified 1,476 shared operational taxonomic units between groups. However, ESCC samples exhibited a greater number of unique ASVs compared to controls, consistent with the observed richness patterns (Figs. 1A–1B). Principal component analysis (PCA) further confirmed distinct microbial community structures between groups, The PERMANOVA analysis revealed a highly significant difference between the two groups (p = 0.00001) (Fig. 1C).

Taxonomic composition

Streptococcus and Neisseria dominated the oral microbiota in both groups. At the genus level, significant compositional differences emerged: Moraxella displayed higher relative abundance in controls (p < 0.001), whereas ESCC samples showed marked enrichment of Gemella (p = 0.0003) and Corynebacterium (p < 0.00001). Conversely, Prevotella_7 was significantly reduced in ESCC patients compared to controls (p = 0.0002) (Fig. 2). The ten microbial genera with the most significant statistical differences are presented in the appendix figure. Among them, five are significantly more abundant in ESCC compared to the control group (Eikenella, Gemella, Dialiste r, F0058), and five are less abundant than the control group (Prevotella_7, Actinomyces, Solobacterium, Campylobacter, TM7x).

Functional profiling

PICRUSt-based metagenomic prediction revealed distinct metabolic pathway enrichment patterns. ESCC-associated microbiota exhibited significant upregulation of genes involved in amino acid metabolism, particularly beta-alanine and valine degradation pathways. In contrast, carbohydrate metabolism pathways, including glucose utilization, showed reduced activity in ESCC samples compared to controls. These findings suggest a metabolic shift toward amino acid catabolism in ESCC-associated microbial communities (Fig. 3).

Figure 3 Functional metabolic pathway alterations in the oral microbiota of ESCC patients.

Discussion

Our research presents a higher α-diversity of oral microbiota in ESCC patients compared to healthy controls. Notably, the Observed, Chao1, and ACE indices demonstrated statistically significant differences (p < 0.05). Comparative analysis identified several microbial genus with significant abundance differences between the cancer and control groups, including Eikenella and Tannerella are consistent with prior studies validating their association with the disease (Bolyen et al., 2019). Furthermore, in advanced ESCC patients undergoing neoadjuvant therapy, responders exhibited significantly higher TM7x abundance compared to non-responders (Douglas et al., 2020). While our study observed higher prevalence of Dialister in ESCC patients, contrasting findings from prior research report its enrichment in healthy controls, highlighting potential context-dependent microbial dynamics (Xue et al., 2023). These findings highlight substantial heterogeneity in the oral microbiota of ESCC patients, likely attributable to multifactorial influences and microenvironment-dependent functional variations among microbial species. To elucidate the precise role of oral microbiota in ESCC pathogenesis, future studies should incorporate larger cohorts, integrated multi-omics approaches (e.g., metagenomics and proteomics), and functional validation experiments.

The intratumor microbiota of esophageal cancer has an important impact on the process, treatment and prognosis of cancer. The data from The Cancer Microbiome Atlas (TCMA) and The Cancer Genome Atlas (TCGA) databases shows the presence of microbiota in esophageal cancer tissues, the abundance of Firmicutes was increased and Proteobacteria was significantly decreased in tumor samples compared to the control group, Wang, et al. discovered a significant presence of Proteobacteria, Negativicutes, and Lactobacillus, which are associated with a positive prognosis, whereas a high abundance of Clostridiales and Fusobacteriales indicates the opposite (Xue et al., 2023; Wang et al., 2021). Fusobacterium nucleatum has been proven to be a sort of carcinogenic microbe in many digestive tract cancers. Researchers have confirmed that intratumoral Fusobacterium nucleatum releases the virulence factor Fn-Dps and binds to ATF3, which disrupts the PD-1 signaling pathway, ultimately inhibiting cancer immunotherapy in ESCC patients (Li et al., 2023). Concurrently, F. nucleatum also exerts an influence on the chemotherapy of EC patients, therefore, it is imperative to analyze the intratumoral microbes in esophageal cancer patients prior to the initiation of treatment (Liu et al., 2021). Among patients with ESCC exhibiting a poor long-term prognosis survival rate, a higher α diversity of microbes and abundance of Lactobacillus were observed, suggestive of a link between microbial composition and the tumor immune microenvironment (Zhang et al., 2023). Patients responding to chemoimmunotherapy exhibited microbial differences, including enrichment of streptococci that positively correlated with the infiltration of GrzB+ and CD8+ T-cells in tumor tissue (Wu et al., 2023). Despite the diverse range of microorganisms present in EC, which exert significant influence on cancer development, management, and outcome, further research is necessary to untangle the complicated mechanisms and spatial patterns of these microbes (Galeano Niño et al., 2022).

The gut microbiota constitutes a crucial component of the human microbiome. However, both the composition and function dysbiosis of their gut microbiota for numerous cancer patients, especially gastrointestinal cancer and esophageal cancer cannot be avoided. There are many studies have reported that gut microbiota differences between EC patients and healthy controls. A study revealed that patients differed from healthy controls in the abundance of Lachnospira, Bacteroides, Streptococcus, and Bifidobacterium. Cancer patients exhibited a decrease in the abundance of bacteria that produce short-chain fatty acids (SCFAs) compared to controls, while the abundance of bacteria that produce lipopolysaccharides (LPS) was elevated (Deng et al., 2021). Another research also observed a higher abundance of the pro-inflammatory bacterium Streptococcus in cancer patients (Cheung et al., 2022). Sasaki et al. (2023) found that the relative abundance of fecal Lactobacillaceae was significantly high for ESCC patients who responded to chemoradiotherapy. Currently, most of the research on the microbiome of esophageal cancer is focused on oral microbiota and cancer tissue samples, while the elaborate mechanisms underlying the interaction between gut microbiota and esophageal cancer remain elusive. Although Pernigoni et al. (2021) have demonstrated that gut microorganisms convert androgen precursors into androgens through metabolism during androgen deprivation therapy for prostate cancer, this process can contribute to drug resistance and consequently diminish the therapeutic effect. However, a significant amount of experimental research is still required to investigate the alterations in gut microbiota during the treatment of esophageal cancer, as well as the underlying mechanisms of these changes.

The oral microbiome plays a significant role in the development and progression of oral squamous cell carcinoma (OSCC) through several molecular mechanisms: firstly, certain oral microorganisms can produce carcinogenic substances such as nitrosamines, sulfides, oxides, and acetaldehyde (Wang et al., 2021). These metabolites can interfere with DNA replication and induce DNA damage and mutations, thereby promoting tumorigenesis. For example, some strains of Candida spp. have high nitrosation potential and can produce nitrosamines that disrupt normal DNA replication (Li et al., 2023). Secondly, the oral microbiome can regulate inflammation and immune responses to facilitate OSCC development. Microbial components, such as lipopolysaccharides (LPS), can activate Toll-like receptors (TLRs) and trigger the release of pro-inflammatory cytokines (e.g., IL-1β, IL-6, and TNF-α). These cytokines promote angiogenesis, cell proliferation, and tumor invasion. Moreover, certain bacteria like Porphyromonas gingivalis and Fusobacterium nucleatum can upregulate immune checkpoint molecules (e.g., PD-L1) and modulate immune cell activity to suppress the host’s immune response, thus facilitating tumor immune evasion (Liu et al., 2021). Additionally, oral microorganisms can promote cell proliferation and inhibit apoptosis by regulating cell cycle proteins and apoptosis-related proteins. For instance, P. gingivalis can accelerate the G1 phase of the cell cycle by upregulating cyclin D1 (Zhang et al., 2023), while F. nucleatum can downregulate the cell cycle-related protein p27, thereby enhancing cell proliferation (Wu et al., 2023). Lastly, the oral microbiome can enhance the invasiveness and metastatic potential of cancer cells by influencing epithelial-mesenchymal transition (EMT). Certain microbes, such as P. gingivalis, can activate signaling pathways like JAK1/STAT3 and recruit tumor-associated neutrophils, promoting OSCC progression (Galeano Niño et al., 2022). Viruses like Epstein-Barr virus (EBV) and human papillomavirus (HPV) can also modulate host cell signaling and immune responses to drive OSCC development (Deng et al., 2021). In summary, the oral microbiome contributes to OSCC through multiple molecular mechanisms, including the production of carcinogens, regulation of inflammation and immune responses, promotion of cell proliferation and anti-apoptosis, and enhancement of cell invasiveness. These mechanisms provide a basis for developing non-invasive diagnostic biomarkers and therapeutic targets for OSCC based on the oral microbiome.

Our study has some limitation include: (1) modest sample size from a single region; While our sample size is modest, it is comparable to pioneering studies in niche populations (Chen et al., 2015; Wang et al., 2019). Future multi-center studies with larger cohorts will validate these findings. (2) 16S rRNA sequencing’s limited functional resolution; (3) lack of mechanistic validation. Future multi-omics studies across diverse cohorts are warranted.

Conclusion

This study uncovers unique oral microbial signatures in ESCC patients from a high-incidence region, characterized by increased richness, taxon-specific shifts, and metabolic reprogramming favoring amino acid catabolism. These findings highlight the potential of microbial biomarkers for ESCC detection and provide mechanistic insights into microbiome-driven carcinogenesis. The geographic specificity of the cohort underscores the urgency of tailored interventions in high-risk populations and advances our understanding of microbial contributions to esophageal cancer.

Supplemental Information

Supplemental Information 1 Top 10 significantly different genus superposed violin graph with confidence intervals in two groups

The left panel displays microbial genera exhibiting significantly higher abundance in the control group relative to the ESCC group, whereas the right panel presents genera that are significantly more abundant in the cancer group compared to healthy controls. Both panels are arranged in descending order based on statistical significance (p-value).

Supplemental Information 2 Raw data for clinical characteristics of enrolled patients and healthy controls

Additional Information and Declarations

Competing Interests

Author Contributions

Human Ethics

Ethics

Field Study Permissions

DNA Deposition

Data Availability

The authors declare there are no competing interests.

Ying Liu conceived and designed the experiments, authored or reviewed drafts of the article, and approved the final draft.

Erman Wu conceived and designed the experiments, prepared figures and/or tables, and approved the final draft.

Fang Cheng performed the experiments, prepared figures and/or tables, and approved the final draft.

Meng Zhang performed the experiments, prepared figures and/or tables, and approved the final draft.

Qian Rou performed the experiments, prepared figures and/or tables, and approved the final draft.

Zinati Nuertai analyzed the data, prepared figures and/or tables, and approved the final draft.

Maorong Xu analyzed the data, prepared figures and/or tables, and approved the final draft.

Shanshan Xu analyzed the data, authored or reviewed drafts of the article, and approved the final draft.

Minghui Li performed the experiments, analyzed the data, prepared figures and/or tables, and approved the final draft.

Lei Zhang performed the experiments, authored or reviewed drafts of the article, and approved the final draft.

Aheli Nasiroula conceived and designed the experiments, authored or reviewed drafts of the article, and approved the final draft.

The following information was supplied relating to ethical approvals (i.e., approving body and any reference numbers):

The Ethics Committee of The Affiliated Cancer Hospital of Xinjiang Medical University (Approval No.: Ethical number: K-2024017) approved this research.

The following information was supplied relating to ethical approvals (i.e., approving body and any reference numbers):

The Ethics Committee of The Affiliated Cancer Hospital of Xinjiang Medical University, K-2024017.

The following information was supplied relating to field study approvals (i.e., approving body and any reference numbers):

Field experiments were approved by the Ethics Committee of The Affiliated Cancer Hospital of Xinjiang Medical university (Approval No.: Ethical number: K-2024017).

The following information was supplied regarding the deposition of DNA sequences:

(Accession: PRJNA1226964).

The following information was supplied regarding data availability:

The data is available at NCBI: PRJNA1226964.

The raw data is available in the Supplemental Files.

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
