# Peer review of "Oropharyngeal microbiome dysbiosis in esophageal squamous cell carcinoma: taxonomic shifts, metabolic reprogramming, and geographic disparities in a high-incidence cohort"

_PeerJ, doi:10.7717/peerj.20009_

## Round 0.1 · original submission · Major Revisions

· Academic Editor

Major Revisions

Please address concerns of both reviewers and amend manuscript accordingly.

·

Basic reporting

No comment

Experimental design

In this manuscript, the authors perform a comparative analysis of the pharyngeal microbiome in adults (a high-risk population) with esophageal squamous cell carcinoma compared to healthy controls. A number of distinctive features of microbiota composition and metabolic pathways were identified.

I have some suggestions for improving this manuscript.

1. It may be appropriate to change the title of the article, since the samples for microbiota analysis were collected from the pharynx.
2. The introduction section would benefit from including some brief information on the mechanisms by which the oral/pharyngeal microbiome may influence cancer development.
3. Line 97. More detailed information about the sample collection could be added (how time has passed since the last meal, etc.).
4. Line 123. Are you certain that the Bray-Curtis distance was used to calculate both alpha and beta diversity?
5. In the Statistical Analysis section, it is worth adding information on how to perform principal component analysis. It is also worth adding more detailed information about the comparison of the 10 most abundant species (as well as metabolic pathways). It is worth adding information about the statistical analysis of the main clinical parameters (presented in the description of Table 1, but not in the Statistical Analysis section). Furthermore, it is also worth indicating which software was used for the analysis.
6. Line 132. The authors have provided significantly more parameters in the text than in Table 1.
7. Line 147: It was noted that the abundance of Moraxella was significantly different between the groups, but this information (indicators) is not included on the graph (it would be worthwhile to include this information on the graph or in the text).
8. The discussion section would benefit from a brief description of the results of this study. It would also be helpful to include the limitations of the study.
9. It would be useful to add the p value to Figure 1C.

Validity of the findings

If possible, it is worth adding repository information (raw 16s rRNA sequencing data) to the text of the article.

Additional comments

No comment

Reviewer 2 ·

Basic reporting

The language of this paper is standardized and normative, with appropriate citation of references. The provided images, tables, and raw data all comply with the required standards.

Experimental design

This study employed 16S rRNA amplicon sequencing to compare the oral microbiome characteristics of ESCC patients and healthy controls in the Northwest region. However, the inclusion of only 30 esophageal squamous cell carcinoma patients may limit the representativeness of the findings due to the small sample size, potentially introducing significant bias in the results. It is recommended to further expand the sample size.

Validity of the findings

As noted in the manuscript, a significant number of studies have already documented the characteristics of the oral microbiome in esophageal squamous cell carcinoma (ESCC). This extensive prior research suggests that the current study may lack novelty, as it does not provide new insights or findings that significantly advance the existing knowledge in this area. Further investigation into unique aspects or mechanisms could enhance the contribution of this research to the field.

Additional comments

The discussion section would benefit from a thorough restructuring to improve both language and logical flow. At present, it dedicates an excessive amount of space to describing studies related to gut microbiota and machine learning, which somewhat detracts from the central focus of the manuscript. It is advisable to shift the emphasis toward a more in-depth exploration of the relationship between the oral microbiome and esophageal squamous cell carcinoma.

---

## Round 0.2 · Minor Revisions

· Academic Editor

Minor Revisions

Please address the remaining concerns of the reviewer and amend the manuscript accordingly.

·

Basic reporting

No comments

Experimental design

The authors have done a great job improving the manuscript.

However, I have a few more minor suggestions.

1. The p value for beta diversity (PCA) should be indicated in Figure 1C and/or the text.

2. "no significant differences in age, gender, and smoking status between the two groups (P > 0.05) as presented in Table 1" is mentioned twice in the text.

3. Some bacteria names are not in italics.

Validity of the findings

I suggest adding the project number to the text of the manuscript.

Additional comments

No comments

---

## Round 0.3 · accepted · Accept

· Academic Editor

Accept

All remaining issues are adequately addressed and the revised manuscript is acceptable now.